# The Utility of Angular Velocity During Back Squat to Predict 1RM and Load–Velocity Profiling

**DOI:** 10.3390/s25196047

**Published:** 2025-10-01

**Authors:** Kyle S. Beyer, Jonathan P. Klee, Jake C. Ojert, Marco D. Grenda, Joshua O. Odebode, Steve A. Rose

**Affiliations:** Resistance Exercise, Physiology, and Sport Laboratory, Health and Exercise Physiology Department, Ursinus College, Collegeville, PA 19426, USA

**Keywords:** velocity-based training, gender difference, linear velocity, angular velocity, inertial measurement unit

## Abstract

Linear velocity is commonly used to estimate 1-repetition maximum (1RM) from a load–velocity profile (LVP), as well as prescribe training intensity. However, no study has assessed angular velocity, which may be more representative of joint motion. The purpose of this study was to compare the prediction of 1RM from linear velocity (1RM_linear_) and angular velocity (1RM_angular_) LVPs in men and women. Fourteen recreationally trained college-aged subjects (7 males, 7 females) completed 1RM testing on day 1, then a randomized submaximal (30–90% 1RM) squat protocol on day 2. Linear velocity was measured with a linear position transducer, while angular velocity was recorded using an accelerometer affixed to the thigh. 1RM_angular_ was not significantly different from actual 1RM (*p* = 0.951), with a trivial effect size (d = 0.02), and nearly perfect correlation with actual 1RM (r = 0.984). 1RM_linear_ had a near perfect correlation with actual 1RM (r = 0.991) but was significantly different than actual 1RM (*p* < 0.001) with a large effect size (d = 1.56). Additionally, 1RM_angular_ had a significantly (*p* = 0.020) lower absolute error (6.7 ± 5.3 kg) than 1RM_linear_ (12.9 ± 8.2 kg). Regardless of prediction method, males (12.9 ± 8.2 kg) had a greater absolute error in 1RM prediction than females (6.7 ± 5.2 kg). During submaximal loads, a significant load × gender interaction was observed for linear velocity (*p* < 0.001), with men showing faster velocities at 30% (*p* = 0.009) and 40% (*p* = 0.044) 1RM, with no significant interaction (*p* = 0.304) of main effect of gender (*p* = 0.116). Angular velocity may provide strength and conditioning coaches a more accurate 1RM prediction during submaximal sets of back squat than using linear velocity; however, neither meet all criteria to be considered highly valid. Lastly, the gender differences in linear velocity at submaximal exercises suggest gender-specific considerations in velocity-based training particularly at lighter loads.

## 1. Introduction

The measurement of barbell velocity during resistance training exercises is a common technique used in strength and conditioning practice typically referred to as velocity-based training (VBT) [1]. VBT techniques typically include establishing an individualized load–velocity profile, tracking progress over time, prescribing resistance training loads, and predicting 1-repetition maximum (1RM) [2,3,4,5,6,7,8,9,10,11,12]. Overall, VBT techniques have been shown to improve athletic performance [13,14,15] and are as effective as percentage-based training [13] or possibly superior [14].

The most common velocity variable measured during VBT techniques is mean velocity. This value represents the average linear velocity of the barbell throughout the concentric phase of the exercise. Mean velocity is preferred to as peak velocity or mean propulsive velocity due to its stronger linear relationship with external load [16] and better agreement between devices [17]. Mean velocity can be measured through a variety of devices including linear position transducers (i.e., GymAware, Tendo) or inertial measurement units (i.e., OUTPUT, Enode) [1]. In general, linear position transducers have greater validity and reliability compared to inertial measurement units [18]. Linear position transducers are attached to the barbell to measure the linear velocity of the barbell; however, it is important to note that the actual joint motion occurring during the exercise is angular motion, which cannot be measured using a linear position transducer. Many inertial measurement units are equipped with a triaxial gyroscope, making them capable of measuring change in angular orientation. Therefore, these devices may be useful in quantifying the angular velocity of specific joints during resistance exercises.

The previous literature has demonstrated that 1RM can be predicted from establishing an individual’s load–velocity profile by measuring mean linear velocity during submaximal loads. Previous research has demonstrated that mean velocity is highly reliable during back squat at various loads [3]. Moreover, mean linear velocity during submaximal loads of back squat have been shown to validly and reliably predict 1RM [4,7,8,9,19]. However, some literature has demonstrated that the accuracy of 1RM estimations from the load–velocity profile is not acceptable [2,20]. Various devices have been shown to accurately predict 1RM from mean linear velocity using linear position transducers [2,7,21], inertial measurement units [21], and smartphone applications [21,22]. However, no study has measured angular velocity at submaximal loads to predict 1RM. Angular velocity may represent an alternative method of predicting 1RM as it is based on the actual joint motion occurring during the exercise. Additionally, gender has been shown to have an effect on the load–linear velocity relationship during bench press and shoulder press [23,24,25,26,27]; therefore, is it important to establish gender differences in joint angular velocity at various loads. Furthermore, nearly all research on the validity of 1RM estimation from load–velocity profiles has been conducted in men [2,4,6,7,9,12,21]. Therefore, assessing the validity of 1RM estimation in females is a current gap in the literature.

The primary purpose of this study was to investigate the utility of measuring angular velocity during barbell back squat in men and women at varying loads to establish individualized load–velocity profiles and predict 1RM. The secondary aim of the study is to examine the gender differences in joint angular velocity during various loads during back squat. It is hypothesized that angular velocity will produce an individualized load–velocity profile that results in a prediction of 1RM that is similar to using linear velocity. Further, it is hypothesized that men will produce a greater angular velocity than women, similar to the previously reported differences in mean linear velocity.

## 2. Materials and Methods

### 2.1. Experimental Design

In this study, subjects reported to the Resistance Exercise, Physiology, and Sport Laboratory in a euhydrated and rested state for two visits separated by at least a week. On the first visit, subjects completed body composition testing, a warm-up, and a 1-repetition maximum (1RM) test on the barbell back squat. On the second visit, subjects completed the same warm-up, followed by seven sets of barbell back squat at 30%, 40%, 50%, 60%, 70%, 80%, and 90% 1RM. Throughout 1RM testing and submaximal loads, subjects had their linear velocity determined via linear position transducer and angular velocity via inertial measurement unit. Overall study design is presented in Figure 1.

### 2.2. Participants

This study was approved by the Ursinus College Institutional Review Board (KB-HEP-0122x) and all subjects signed an informed consent prior to beginning the study. A total of 14 (M = 7, F = 7) recreationally trained men (22.0 ± 0.7 y, 177.5 ± 6.7 cm, 91.8 ± 14.5 kg) and women (20.5 ± 1.2 y, 163.4 ± 5.3 cm, 65.0 ± 5.8 kg) were recruited to complete this study. All subjects were free from injury and had completed resistance training at least twice per week for the last six months. Furthermore, all subjects were familiar with the barbell back squat and demonstrated proper and consistent technique throughout the study as determined by a Certified Strength and Conditioning Specialist (CSCS).

### 2.3. Body Composition

Subjects were first assessed for height and body mass using a stadiometer and digital scale (seca, Chino, CA, USA), respectively. After, body composition was assessed via bioelectrical impedance spectroscopy (SFB7, Impedimed, Carlsbad, CA, USA). Subjects were required to lay supine for 5 min to allow for fluid shifts. Two single-tab electrodes were placed 5 cm apart on the dorsal surface of the wrist and ankle on the right side of the body. The BIS device measured whole body water and extracellular fluid based on Cole modelling with Hanai mixture theory, which was then used to calculate percent body fat (Men: 17.0 ± 6.3%; Women: 22.8 ± 3.6%).

### 2.4. Warm-Up

Prior to exercising, all subjects completed a general and specific warm-up. The general warm-up consisted of riding a cycle ergometer for 5 min at the participant’s preferred resistance. The specific warm-up consisted of 10 body weight squats, 10 alternating lunges, 10 walking knee hugs, and 10 walking butt kicks.

### 2.5. 1-Repetition Maximum Testing

On the first visit, all subjects completed a 1RM test of the barbell back squat. After completing the general and specific warm-up, subjects completed three warm-up sets using an estimated resistance of 50%, 70%, and 90% of their perceived 1RM at 8, 5, and 1 repetitions, respectively. After, subjects began their attempts at a 1RM. After each successful attempt, the load was increased and another set attempted, while after failed attempts, the load was reduced prior to attempting another set. All subjects were provided five attempts to achieve their 1RM. A 3–5-min rest period was provided between each attempt. All 1RM testing was supervised by a CSCS who determined successful and failed attempts, as well as load increases and decreases. The highest load lifted with proper technique was considered their 1RM (1RM_actual_).

### 2.6. Submaximal Testing

On the second visit, subjects completed the same general and specific warm-up followed by seven submaximal sets of the barbell back squat exercise at loads of be 30%, 40%, 50%, 60%, 70%, 80%, and 90% of the previously determined 1RM. The order of the sets was randomly determined for each participant, with the exception that 80% and 90% could not occur first or second. The sets at 30–70% required the participant to complete five repetitions, while the sets at 80% and 90% required the subject to complete three repetitions. A 3–5-min rest period was provided between each set. All submaximal testing was supervised by a CSCS who ensured repetitions were completed to their full range of motion. For each repetition subjects were instructed to complete the repetition with maximal intent and to move as fast as possible.

### 2.7. Linear and Angular Velocity Measurements

During the 1RM and submaximal testing, average linear velocity and angular velocity were measured for each repetition. Linear velocity was measured using a linear position transducer (Tendo Weightlifting Analyzer, TENDO Sports Machines, Trenchin, Slovak Republic) positioned directly beneath the right collar of the barbell with a retractable cord attached to the collar. The linear position transducer measured the change in displacement of the barbell (m) throughout the repetition over the time of the repetition (s). Angular velocity was measured using an inertial measurement unit (Output Sports, Dublin, Ireland) attached to the lateral aspect of the thigh at 2/3rd the distance between the greater trochanter and the lateral condyle of the femur. The inertial measurement unit measured the change in orientation (°) throughout the repetition over the time of the repetition (s). Both the linear position transducer [28,29] and inertial measurement unit [30] have demonstrated acceptable test–retest reliability. For both linear and angular velocity, the average of the entire concentric portion of each repetition was determined. For each set, the average of the two closest repetitions were chosen for analysis. Figure 2 presents the setup of the linear and angular velocity measurements during testing. 

### 2.8. 1-Repetition Maximum Estimation

Load and velocity data from the submaximal testing were used to estimate 1RM using linear (1RM_linear_) and angular (1RM_angular_) velocities. First, the seven submaximal loads and velocities were plotted. Then, the slope and intercept of the line of best fit were determined. Finally, the estimated 1RM was calculated by using the slope, intercept, and velocity data from the participants 1RM_actual_. This process was completed for linear and angular velocities. Figure 3 presents the 1RM estimation methodology for linear velocity and angular velocity using an example subject’s data. 

### 2.9. Statistical Analysis

All linear velocity and angular velocity () data were determined to be normally distributed via Shapiro–Wilk test. Accuracy of 1RM_linear_ and 1RM_angular_ compared to 1RM_actual_ was evaluated with paired samples *t*-test, Cohen’s d effect size, Pearson correlation coefficient, and standard error of the estimate. The magnitude of effect size was interpreted as trivial (<0.2), small (0.2–0.59), moderate (0.60–1.19), large (1.2–2.0), and very large (>2.0) [31]. The magnitude of the correlation coefficient was interpreted as trivial (<0.1), small (0.1–0.29), moderate (0.3–0.49), large (0.50–0.69), very large (0.70–0.89), and nearly perfect (>0.90) [31]. In accordance with previous research [21], the 1RM predictions were considered highly valid if there was no significant difference when compare to the actual 1RM, the effect size was trivial or small, the correlation coefficient was nearly perfect, and the standard error of the estimate was <5 kg. Furthermore, to examine the error in the two 1RM predictions, absolute error between 1RM_actual_ and each predicted 1RM was calculated and analyzed with gender × method mixed factorial ANOVA. Additionally, Bland–Altman plots were generated to display the agreement (bias and 95% limits of agreement) between each predicted 1RM and 1RM_actual_. Lastly, gender differences in linear and angular velocity at submaximal loads and 1RM were assessed via gender × load mixed factorial ANOVA. Any significant interactions were followed with Tukey post hoc comparisons. Alpha levels were set to *p* < 0.05. All data are presented as mean ± standard deviation.

## 3. Results

In terms of accuracy, 1RM_angular_ met three out of four criteria to be considered highly valid, while 1RM_linear_ only met one out of four criteria. 1RM_linear_ experienced a significant difference when compared to 1RM_actual_, had a large effect size, and a standard error of the estimate greater than 5 kg; however, the correlation between 1RM_linear_ and 1RM_actual_ was nearly perfect. For 1RM_angular_, there was no significant difference when compared to 1RM_actual_, a trivial effect size, and a nearly perfect correlation to 1RM_actual_; however, the standard error of the estimate was still greater than 5 kg. The accuracy values are presented in Table 1. In terms of absolute error, there was no significant gender × method interaction (*p* = 0.420); however, there was a significant main effect of gender (*p* = 0.025) and main effect of method (*p* = 0.020). Regardless of prediction method, males (12.9 ± 8.2 kg) had a greater absolute error in 1RM prediction than females (6.7 ± 5.2 kg). Furthermore, 1RM_linear_ (12.9 ± 8.2 kg) had a significantly greater absolute error than 1RM_angular_ (6.7 ± 5.3 kg). All 1RM data are presented in Table 2 with Bland–Altman plots presented in Figure 4.

ANOVA revealed a significant gender × load interaction (*p* < 0.001) for linear velocity. Post hoc tests revealed a significant difference in linear velocity between males and females at 30% (*p* = 0.009) and 40% (*p* = 0.044) of 1RM. No other differences between genders were noted. For angular velocity, ANOVA revealed no significant gender × load interaction (*p* = 0.304) or main effect of gender (*p* = 0.116). Linear and angular velocity data between genders are presented in Figure 5.

## 4. Discussion

The purpose of this study was to assess the accuracy of predicting back squat 1RM from angular velocity, as well as measure sex differences in angular velocity. The findings of this study indicate that 1RM_angular_ is a more accurate prediction than 1RM_linear_ with less absolute error; however, neither prediction met all criteria to be considered highly valid. As for gender differences, males and females produced significantly different average linear velocities during barbell back squat at loads of 30% and 40% of 1RM with no differences at higher loads; however, no differences were noted between genders for angular velocity at any load.

The current results show that 1RM predicted from linear mean velocity at submaximal loads did not produce an accurate prediction. Despite a nearly perfect correlation between 1RM_linear_ and 1RM_actual_, these values were statistically different with a large magnitude of difference. The prediction from linear velocity results in approximately a 12.9 kg overestimation in 1RM with all subjects being overpredicted. These results are similar to findings from previous research on back squat and deadlift 1RM prediction [2,32]. The accuracy of the prediction of 1RM from the individual load–velocity profile may be affected by a number of factors, including the number of loads used, which loads are used, the minimum velocity threshold, the velocity measuring device, and the exercise equipment [5,33]. However, studies have demonstrated that the number of loads included in the model or the minimum velocity threshold used have limited impact on the accuracy of the prediction equation [6,20,34]. A larger factor to consider is that many studies investigating the ability to predict 1RM from submaximal barbell velocity have typically used a Smith Machine [6,21], whereas the current study used free weights. While the Smith Machine has advantages in controlling non-vertical movement, there is a limitation in the practical application for strength and conditioning professionals.

Unlike 1RM_linear_, 1RM_angular_ was not significantly different from 1RM_actual_ with nearly perfect correlation between variables. Additionally, the absolute error of 1RM_angular_ was significantly lower than 1RM_linear_. However, the standard error of the estimate for 1RM angular was still larger than the criteria to be considered highly valid, so caution should be used when predicting back squat 1RM from angular velocity. Moreover, this is the first study to measure angular velocity during back squat, so more research must be conducted to corroborate these findings. Additionally, it is important to note that while the current study did measure angular velocity as the change in orientation of the IMU affixed to the thigh, it does not represent a specific joint angular velocity during the back squat. Future research should consider measurement of each individual joint to see how their angular velocity differs during a back squat.

The reported gender differences in linear velocity were only noted at loads of 30 and 40%, which have been previously referred to as the “Speed/Strength” zone [35]. Interestingly, this zone has been associated with linear velocities ranging from 1.0 to 1.3 m/s. In the current study, female subjects were on average slower than these previously reported values. It is possible that the previously established velocities were based on data from predominantly male subjects. Caution should be used by strength and conditioning professionals when applying established norms to all individuals when the source of the normative data is likely from a homogenous group. Other studies have reported similar differences in linear barbell velocity between men and women during military press and bench press [23,24,25]. Considering the previously reported gender differences in upper body exercises and the gender differences in back squat of the current study, it is advisable for strength and conditioning professionals to establish individualized load–velocity profiles prior to implementing velocity-based training. Interestingly, no differences were noted between genders in angular velocity at any load, which may indicate it is less impacted by the physiological differences between sexes. However, more research on angular velocity should be conducted to further elucidate any effect of gender on angular velocity during resistance exercise. Another factor to consider in the discrepancy between gender differences in linear velocity and angular velocity is any potential differences in limb length between genders, which should be included in future investigations.

This current study does have some limitations that limit the generalizability of the findings. Firstly, the study sample size is limited to 14 recreationally trained men and women. Future studies should include a larger sample size to corroborate the findings of the present manuscript. Additionally, investigating the utility of angular velocity in a highly trained group of individuals would be important to apply the findings in that population. Another limitation of the current study is that the measurement of angular velocity by the IMU is not a direct measurement of joint motion, but rather thigh rotation velocity. Measurement of specific joint velocity during squat could assist in the prediction of 1RM; however, it would be necessary to consider all involved joints in that analysis, which could limit practicality.

## 5. Conclusions

Measuring angular velocity to predict 1RM may be a more accurate method of estimation than linear velocity when strength and conditioning professionals choose to estimate 1RM; however, 1RM prediction with angular velocity did not meet all criteria to be considered highly valid. While measurement of angular velocity during resistance exercise is not as common in practice, it may represent a useful tool for practitioners when looking to estimate maximal strength during back squat. Furthermore, the prediction of 1RM appears to have a greater error in men than women regardless of method, thus strength and conditioning coaches should use caution when attempting to estimate 1RM in men from barbell velocity. More studies should be conducted on the use of angular velocity within resistance exercise, including different exercises, devices, and over longer training programs.

## Figures and Tables

**Figure 1 sensors-25-06047-f001:**
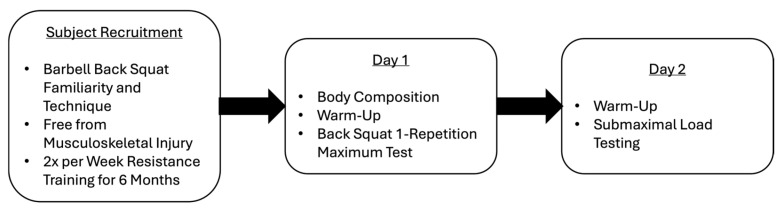
Study design. Linear and angular velocities measured during 1-repetition maximum and submaximal load testing.

**Figure 2 sensors-25-06047-f002:**
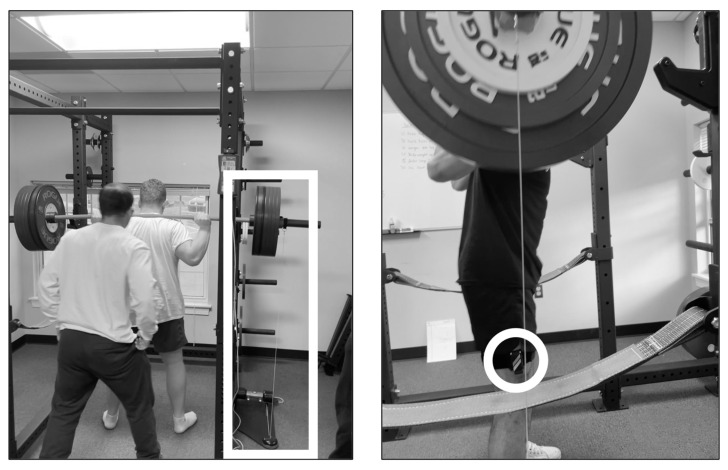
Depiction of the setup of the linear position transducer (left picture, white rectangle) and inertial measurement unit (right picture, white circle).

**Figure 3 sensors-25-06047-f003:**
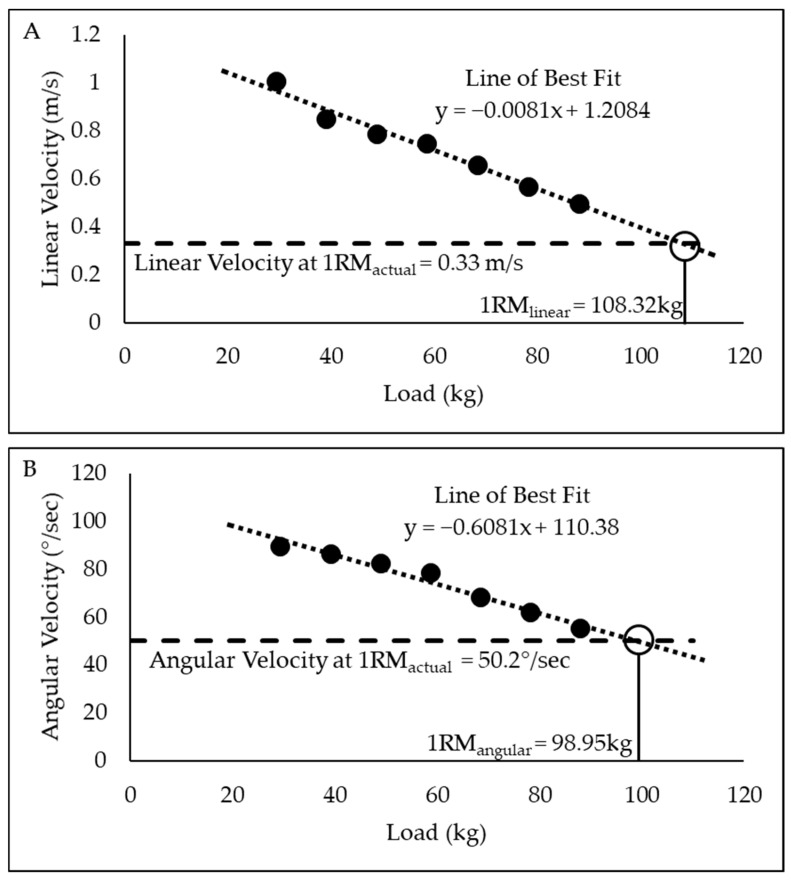
Estimation of 1-repetition maximum (1RM) from linear velocity (**A**) and angular velocity (**B**) using example subject data. 1RM_actual_ for this subject was 97.7 kg.

**Figure 4 sensors-25-06047-f004:**
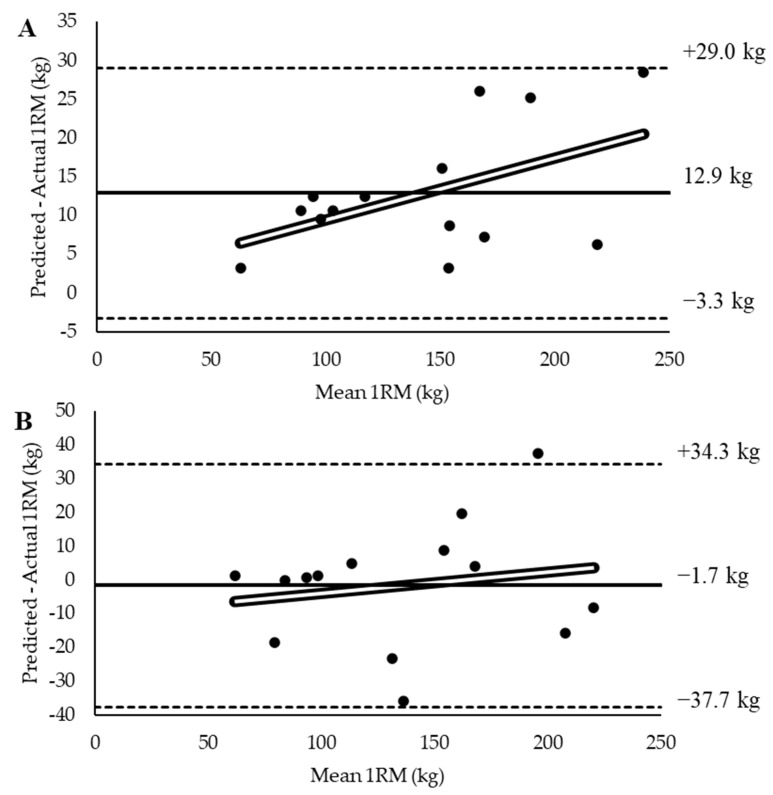
Bland–Altman plots showing the comparison between actual 1RM and (**A**) linear velocity-predicted 1RM and (**B**) angular velocity-predicted 1RM. Solid line represents the mean difference between variables with horizontal dashed lines representing the 95% confidence interval of the mean difference. Unfilled line represents the line of best fit.

**Figure 5 sensors-25-06047-f005:**
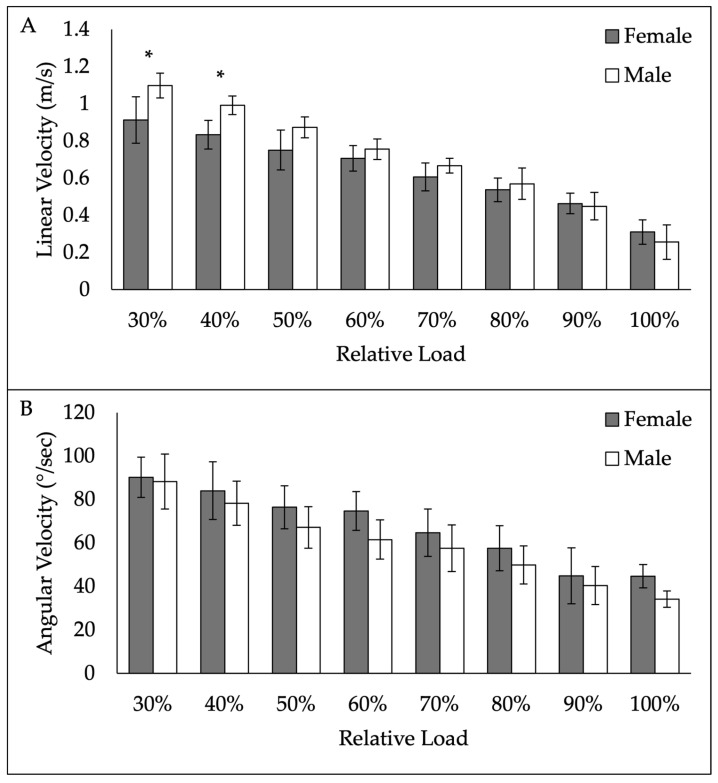
Gender differences in linear velocity (**A**) and angular velocity (**B**) from 30% to 100% 1-repetition maximum. * denotes significant difference between genders.

**Table 1 sensors-25-06047-t001:** Comparison of the 1RM_actual_ versus the predicted 1RM from linear velocity (1RM_linear_) and angular velocity (1RM_angular_) for the whole study sample.

Predicted 1RM	Mean ± SD (kg)	*p* Value	Cohen’s d (95% CI)	r (95% CI)	SEE (95% CI)
1RM_linear_	149.79 ± 53.75	<0.001	1.56 (0.76–2.34)	0.991 (0.969–0.997)	16.33 (11.71–26.95)
1RM_angular_	137.07 ± 48.41	0.951	0.02 (−0.51–0.54)	0.984 (0.950–0.995)	9.11 (6.53–15.04)

1RM, 1-repetition maximum; SD, standard deviation; *p* value, dependent *t*-test *p* value; 95% CI, 95% confidence interval; r, Pearson correlation coefficient; SEE, standard error of the estimate. 1RM_actual_ = 136.92 ± 49.61 kg.

**Table 2 sensors-25-06047-t002:** Actual and predicted 1-repetition maximums (1RM) for males and females with absolute errors between actual and predicted 1RMs. Data presented as mean ± standard deviation.

Gender	1RM_actual_ (kg)	1RM_linear_ (kg)	Abs. Error 1RM_linear_ (kg)	1RM_angular_ (kg)	Abs. Error 1RM_angular_ (kg)
Male	176.91 ± 30.98	191.94 ± 35.80	15.02 ± 10.97	177.54 ± 25.65	10.82 ± 4.00
Female	96.93 ± 25.32	107.65 ± 28.88	10.72 ± 3.94	96.59 ± 24.42	2.68 ± 2.26

## Data Availability

All data will be made available upon reasonable request.

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
