# Peer review of "The Utility of Angular Velocity During Back Squat to Predict 1RM and Load–Velocity Profiling"

_sensors, 2025, doi:10.3390/s25196047_

Round 1

Reviewer 1 Report

Comments and Suggestions for Authors

Dear authors:

The publication recommendation is Major Revisions. While the research question is pertinent and the findings offer interesting observations, the manuscript requires substantial modifications to address critical methodological deficiencies, clarify the terminology used, and temper certain conclusions in light of the study's limitations. These revisions are essential to ensure scientific rigor, accuracy, and proper interpretation of the presented data.

Reviewer 2 Report

Comments and Suggestions for Authors
  1. Abstract and introduction – poor motivation for the study, try to make methods compartment in the abstract shorter.
  2. General idea of the differences in linear and angular velocities – normally linear velocity is a product of angular velocity and radii, so significant difference between behavior of this quantities mean substantial dispersion of body length (or other quantity – picture should be given).
  3. Line 49 ’is it’- question form, I think this is mistype, please correct this phrase.
  4. Chapter 2.7 methods strictly need graphical support,
  5. Error of measurements for all units should be given,
  6. The sentence in 145-146 undermine some formulae – it should be provided in the text
  7. Pearson test undermine normality of the data – which criterion was used to value this feature of the sample? Which anthropomorphic parameters distributed normally and which are not?
  8. In the discussion-conclusion compartment authors should answer the question – how their result can be used by the community?

Reviewer 3 Report

Comments and Suggestions for Authors

The results section needs to be rewritten, introducing more figures and fewer tables for a better understanding of the results.

Without a good results section, the manuscript cannot be accepted, otherwise the whole discussion section needs to be revised and clearly state why both speeds are recommended, what practical applications are proposed and the real possibility of measuring both speeds in a real-use environment with athletes.

Other limitations are:
Very small sample size (n=14): Low statistical power. Difficult to generalise results.

Subjects:  Recreationally trained subjects, not elite athletes or clinical population.

Although angular velocity improves estimation, it does not meet the criterion of SEE <5 kg, limiting its applicability as a reference method.

Lack of comparison with more advanced prediction models (non-linear regression, machine learning).

Reviewer 4 Report

Comments and Suggestions for Authors

Review of the paper (manuscript ID sensors-3753440)

Title: The Utility of Angular Velocity During Back Squat To Predict 1RM And Load Velocity Profiling

Authors: Kyle S. Beyer, Jonathan P. Klee, Jake C. Ojert, Marco D. Grenda, Joshua O. Odebode and Steve A. Rose

The article submitted for review provides an interesting resource for future readers concerned with the practice of increasing strength training effectiveness and the link between linear and angular velocity parameters for estimating future strength training effectiveness. Nevertheless, the article, in my opinion, requires corrections and additions. I have included a list of critical comments below:

  1. The cited references in the Introduction section should be described in more detail so that the reader can assess the contribution of these papers to the scientific topic being addressed. Hence, citations [2-12] for example should be described in more detail.
  2. It is useful to use a flow chart to describe the research methodology adopted and then describe its individual elements in the following sections.
  3. The article lacks, in my opinion, illustrations showing the nuances of the research conducted with the selected study group. It would have been worthwhile to highlight the measurements, the apparatus and its location, statistical and methodological errors, the analysis and processing of information and their uncertainty. This is what I feel is missing from the text. The authors only mention selected names of the devices used and do not refer to the method of measurement and its uncertainty or the reasons for choosing such solutions. In a Sensors journal, I would have expected a broader and more detailed description of the technical aspects.
  4. In the section 3 - I feel unsatisfied with the results presented, in particular with Table 3. The analysis and discussion of these results should be more detailed.
  5. The section Conclusion should clarify and elaborate on the results achieved, the limitations of the method of measuring and estimating the linear and angular velocity parameters and should specify directions for future research.

Round 2

Reviewer 2 Report

Comments and Suggestions for Authors

After the revision by authors I have to highlight:

  1. Please split sentence in lines 70-72.
  2. Sec 2.9 Shapiro-Wilk test value should be given
  3. I recommend to add actual references (only 6 for the last 5 years)

The rest issues are clear now, pictures and tables are good.  

Author Response

  1. Please split sentence in lines 70-72.
    • We have made this revision in the updated version of the manuscript. 
  2. Sec 2.9 Shapiro-Wilk test value should be given
    • Please see the W statistics for all data in the table below. We have chosen to include in the manuscript that all linear velocity data were W>0.919 and for angular velocity W>0.894. If you believe there would be value in including every W statistic in the manuscript, we can make this adjustment.
    • Descriptive Statistics    
        AV30 AV40 AV50 AV60 AV70 AV80 AV90 30 Angular 40 Angular 50 Angular 60 Angular 70 Angular 80 Angular 90 Angular 1RM_Load_kg 1RM_AV 1RM Angular
      Shapiro-Wilk   0.937   0.953   0.919   0.987   0.924   0.937   0.983   0.962   0.894   0.964   0.968   0.950   0.969   0.942   0.950   0.985   0.958  
      P-value of Shapiro-Wilk   0.376   0.605   0.212   0.997   0.254   0.377   0.987   0.754   0.091   0.782   0.852   0.568   0.864   0.447   0.562   0.994   0.691  

  3. I recommend to add actual references (only 6 for the last 5 years)
    • We have added a few references from within the last few years to enhance the previous literature sources. If there are any specific statements you would like to see additional references, please let us know. 

Reviewer 3 Report

Comments and Suggestions for Authors

All my concerns have been answered, I recommended to accept the manuscript.

Author Response

Thank you for all of your comments.

Reviewer 4 Report

Comments and Suggestions for Authors

Review of the paper (manuscript ID sensors-3753440) - the second round

Title: The Utility of Angular Velocity During Back Squat To Predict 1RM And Load Velocity Profiling

Authors: Kyle S. Beyer, Jonathan P. Klee, Jake C. Ojert, Marco D. Grenda, Joshua O. Odebode and Steve A. Rose

Dear Authors,

In my opinion, the revisions introduced by the authors in response to the first reviewer's critique are insufficient to persuade me to issue a final positive recommendation for further processing of the article. Nevertheless, I suggest that the manuscript be thoroughly re-edited, as I am confident it would gain recognition within a different publication pathway. The topic is undoubtedly interesting; however, I believe the authors should make a greater effort to ensure the manuscript meets the scholarly standards required by the Sensors MDPI Journal.

Author Response

In my opinion, the revisions introduced by the authors in response to the first reviewer's critique are insufficient to persuade me to issue a final positive recommendation for further processing of the article. Nevertheless, I suggest that the manuscript be thoroughly re-edited, as I am confident it would gain recognition within a different publication pathway. The topic is undoubtedly interesting; however, I believe the authors should make a greater effort to ensure the manuscript meets the scholarly standards required by the Sensors MDPI Journal.

We appreciate your comments and feedback on this manuscript from your previous round of comments. We have added the study design figure you previously recommended under the study design section. We have made substantial revisions from the first round of revisions to every aspect of the manuscript, and we would be willing to make more revisions if there were more specific comments that you felt would enhance the manuscript. Thank you for your time and consideration of our manuscript.